# Reversible Upregulation of the Senescence-Associated Beta-Galactosidase Marker Induced by Cell Detachment in Cancer Cells

**DOI:** 10.3390/cells14211667

**Published:** 2025-10-24

**Authors:** Nina Semenova, Juan Sebastian Yakisich, Robyn Ascue, Anand K. V. Iyer, Neelam Azad

**Affiliations:** 1Department of Pharmaceutical Sciences, School of Pharmacy, Hampton University, Hampton, VA 23668, USA; robyn.ayscue@hamptonu.edu; 2School of Pharmacy, Hampton University, Hampton, VA 23668, USA; anand.iyer@hamptonu.edu; 3The Office of the Vice President for Research, Hampton University, Hampton, VA 23668, USA; neelam.azad@hamptonu.edu

**Keywords:** senescence, quiescence, β-galactosidase, spheroids, plasticity, galactose

## Abstract

During metastasis, cancer cells detach from the primary tumor, and the floating cells enter the circulation and reattach in distant organs. Floating cells are highly chemoresistant to anticancer drugs, but the underlying mechanisms are poorly understood. We hypothesized that floating cells transition into a quiescent/senescent (Q/S) state. Using human lung carcinoma H460 and H23, human prostate adenocarcinoma PC3, and human breast adenocarcinoma MDA-MB-231 cells, we found (1) a progressive increase in activity of β-galactosidase (β-Gal), a marker associated with Q/S cells, (2) a transition to a non-proliferative state while growing under anchorage-independent conditions, and (3) upon reattachment, the β-Gal activity returned to the basal level and cells resumed proliferation. Similar experiments were performed in parallel with cells treated with etoposide (Eto), a well-known inductor of senescence. Eto-untreated floating cells resumed proliferation faster and showed a quicker decrease in β-Gal activity compared to Eto-induced senescent cells. We conclude that cell detachment per se triggers a reversible (plastic) increase in β-Gal. Our findings provide a partial explanation for chemoresistance under anchorage-independent conditions and a new target to eliminate highly resistant floating cells. Ultimately, eliminating Q/S floating cells may prevent or reduce metastasis.

## 1. Introduction

Microenvironmental conditions have profound effects on the sensitivity of cancer cells to anticancer drugs [1,2,3,4]. Early studies using cancer cells grown in different microenvironments revealed that cancer cells grown under anchorage-independent conditions (AICs) tend to aggregate and form complex 3D structures, usually called spheroids or tumorspheres, and become highly resistant to conventional anticancer drugs [5,6,7,8,9]. It was initially thought that the increased chemoresistance was the result of the complexity of the 3D structure, which may limit the access to drugs or form metabolic and physiological gradient [10]. Using MDA-MB-231 breast and PC3 prostate cancer cell lines that do not form spheroids under AICs, we demonstrated that floating conditions per se increase the chemoresistance of cancer cells. This result was confirmed by blocking spheroid formation with anti-cadherin antibodies in [10].

Chemoresistance can develop through many mechanisms, including induction of quiescence [11] and senescence [12]. Quiescent cancer cells (QCCs) are nonproliferating cells arrested in the G0 phase, characterized by ki67^low^ and p27^high^ [11]. In non-cancer cells, cellular senescence is an irreversible form of long-term cell-cycle arrest, caused by excessive intracellular or extracellular stress or damage [13]. However, escape of the senescence state has been observed in cancer cells and linked to tumor chemoresistance, tumor progression, and metastasis [14,15]. Interestingly, both quiescent and senescent cells are associated with increased expression of the senescence-associated beta galactosidase (SA-βG) marker [16].

The fate of cells detached from their extracellular matrix is a regulated process: cell detachment in normal cells leads to a form of programmed cell death called “anoikis” that prevents colonization of normal cells in other tissues. On the contrary, detached cancer cells survive anoikis. This property is a key step in the process of metastasis. Kim et al. demonstrated that senescent cells are more resistant to anoikis [17].

In this article, we explore the hypothesis that cell detachment per se induces a reversible quiescent/senescent-like (Q/S-like) state that may contribute to the chemoresistance of cancer cells growing under AICs.

## 2. Materials and Methods

Cell lines: Human lung carcinoma H460, human lung adenocarcinoma H23, human prostate adenocarcinoma PC3, and human breast adenocarcinoma MDA-MB-231 cell lines were purchased from the American Type Culture Collection (Manassas, VA, USA, www.atcc.org accessed on 22 March 2025) and routinely cultured in complete media (CM) consisting of RPMI-1640 medium (Sytiva; Logan, UT, USA) supplemented with 10% fetal bovine serum (FBS) (Sytiva; Logan, UT, USA), 2 mM L-glutamine, 100 U/mL penicillin, and 100 mg/mL streptomycin (all from Mediatech; Manassas, VA, USA). To create AICs, the cells were grown to 60–70% confluency in 10 cm Petri dishes, trypsinized, and resuspended in CM in ultra-low attachment plates (ULAPs). Media were changed twice a week. All cells were maintained in a 5% CO_2_ environment at 37 °C.

Reagents and Drug Preparation: X-Gal was purchased from MP Biomedicals (Solon, OH, USA). Eto was acquired from Millipore (Burlington, MA, USA), prepared as stock solutions (10 mM) in dimethyl sulfoxide (DMSO), and stored at −20 °C. Stock solutions were freshly diluted to the appropriate concentration with cell culture medium before use. The control experiment contained the highest concentration (<1%) of DMSO only.

Senescence assay: β-Gal expression was assessed by X-gal staining as described before [18]. Adherent cells were washed twice with PBS and then fixed with 3.7% formaldehyde diluted in PBS for 30 min at room temperature. The cells were washed again with PBS and incubated overnight at 37 °C (without CO_2_) with X-gal staining solution that consisted of 1 mg/mL of 5-bromo-4-chloro-3-indolyl b-Dgalactoside, 40 mM citric acid sodium phosphate (pH 6.0), 5 mM potassium ferrocyanide, 5 mM potassium ferricyanide, 150 mM NaCl, and 2 mM MgCl2. SA-βG-stained cells were visualized using an Axio Observer microscope (Carl Zeiss Microscopy GmbH; Gottingen, Germany) after 18–24 h of incubation. SA-βG staining of floating cells and spheroids was performed using the same procedure with the exception that all washes were performed after light centrifugation (1500 rpm, 5 min). Aliquots of the cell suspension pellet (1× *g*) were loaded in slides, and images were obtained using an Axio Observer microscope (Carl Zeiss Microscopy GmbH; Gottingen, Germany).

Colony-forming assay: PC3 and MDA-MB-231 cells were grown to 60–70% confluency, trypsinized, and placed into ULAPs in CM containing either 50 μM Eto or DMSO (control). Media were changed on day 3–4. After 7 days under AICs, the cells were counted, and an equal number of viable cells were plated in 6-well plates (1000 cells per well for MDA-MB-231 and 500 cells per well for PC3). The cells were allowed to grow for 11 days, and then they were washed twice with PBS and fixed with 3.7% formaldehyde diluted in PBS for 30 min at room temperature. The cells were washed again with PBS and stained with 0.01% of crystal violet in dd H_2_O.

Proliferation assay: The cell proliferation rate was determined by quantification of viable cells using the trypan blue exclusion method [19].

Viability assay: PC3 and MDA-MB-231 cells were grown to 60–70% confluency, trypsinized, and placed into ULAPs in CM containing either 50 μM Eto or DMSO (control). Media were changed on day 3–4. After 7 days under AICs, the cells were counted and seeded at a low density into 35 mm Petri dishes for re-attachment. Media were changed on days 3 and 5. On day 7 after seeding, the cells were incubated with MTT (3-(4,5-di methyl thiazol-2-yl)-2,5-diphenyltetrazolium bromide; 0.5 mg/mL) for 2 h. Images were obtained using an Axio Observer microscope (Carl Zeiss Microscopy GmbH; Gottingen, Germany). This assay was used as a qualitative method to evaluate the metabolic activity of Q/S cells, as shown in Appendix A.

Bioinformatic and statistical analysis: A list of 306 unique gene names associated with human carbohydrate metabolism pathways was obtained from the Kegg pathway database [www.kegg.jp; last accessed on 22 March 2025]. Three additional genes associated with the human Leloir pathway were selected from Sharpe et al. [20]. hg38 genome-aligned breast cancer gene expression (RNA-seq mRNA) data for 58,006 unique genes and corresponding patient (clinical) data were pulled from the TCGA [https://www.cancer.gov/ccg/research/genome-sequencing/tcga; last accessed on 22 March 2025] database using custom R scripts with the TCGAbiolinks [21,22] R package (version 3.20) from Bioconductor [23]. The transcriptomic data consisted of 113 normal and 1111 primary tumor solid breast cancer (BRCA) tissue samples. The downloaded gene expression data was filtered to remove any genes with an insufficient read count and then normalized using the trimmed mean of M-values (TMM) method [24] via the R function VOOM [25], to help ensure that differences in gene expression levels were due to biological reasons and not laboratory artifacts or variations in library size. A differential gene expression analysis (DGE) was performed on the normalized data using the R package limma [25,26,27] to assess the degree of difference for a given gene’s expression levels in breast cancer tumor tissue compared to the same gene’s expression levels in normal breast tissue. We limited the DGE output to only the 1000 of the topmost differentially expressed genes with *p*-values less than 0.05. From these 1000 genes, we extracted DGE data for any gene whose name was in the list of 309 carbohydrate metabolism pathway genes or Leloir pathway genes and retained them for survival probability analysis. To determine how expression levels of the topmost differentially expressed carbohydrate metabolism pathway/Leloir pathway genes affect breast cancer patient prognosis, we used the median expression level of each gene to identify which patients over-expressed the genes and which patients under-expressed them. We created Kaplan–Meier (KM) curves from this data using the R functions survfit from the survival R package (https://cran.r-project.org/web/packages/survival/vignettes/survival.pdf; last accessed on 22 March 2025) [28] and ggsurvplot from the survminer R package [https://cran.r-project.org/web/packages/survminer/survminer.pdf; last accessed on 22 March 2025], to examine changes in patient survival probability as a function of time. The same data used to develop the KM plots was used to correlate patient survival with gene expression level. A *p*-value cutoff of 0.05 was used to determine the statistical significance of each gene’s expression correlation with survival.

## 3. Results and Discussion

### 3.1. Floating Spheroids Growing Under AICs Display High Levels of SA-βG Activity

Cancer cells when growing under AICs tend to form three-dimensional structures (3D), usually called spheroids or tumorspheres, that show high resistance to conventional anticancer drugs. To test the hypothesis that a Q/S status occurs in floating spheroids, H460 and H23 cells were cultured under AICs for 15 days. Cell aliquots were taken at days 1, 3, 5, 7, and 15. The cells were fixed and stained to analyze the expression of SA-βG. As a control, the same cells were plated at a low density in Petri dishes, then allowed to attach and grow under anchorage-dependent conditions (ADCs) for 7 days. Figure 1 shows that after 7 days, the majority of floating spheroids developed blue staining (indicating a high SA-βG activity). In contrast, the intensity of the blue signal in cells growing under ADCs for the same amount of time was negligible: a small fraction of attached cells was blue. The lack of blue color in floating cells on day 1 followed by the strong blue signal detected on day 7 is not a technical artifact but a clear indication that the SA-βG activity increases over time under AICs.

Appendix A shows a clearer progression of the increase in the SA-βG activity over time (1, 3, 5, and 7 days) in H460 and H23 cells. While the picture shown on day one looks like H460 and H23 cells form spheroids, these structures are not yet true spheroids and can be mechanically separated by gently aspirating and expelling with a pipette. However, on day 7, these clumps strongly adhere to each other, forming spheroids (although the shapes are not fully spherical) that cannot be mechanically separated. This is the reason we called these cell lines “spheroid-forming” cancer cell lines.

It is worth mentioning that increased SA-βG activity has been observed in the MDA-MB-231 cell culture, forming localized 3D cell clusters in a confluent 2D tumor layer [29], but to the best of our knowledge, not in floating cells.

### 3.2. SA-βG Activity Increases in Floating Cells Independently of Their Ability to Form Complex 3D Structures

To rule out the possibility that the increased SA-βG activity was the result of the complex 3D architecture, we repeated these experiments in two additional cell lines, MDA-MB-231 and PC3, that naturally do not form spheroids when growing under AICs. Instead, they form loose aggregates that are still highly resistant to conventional anticancer drugs [10]. In this study, we took advantage of this property of MDA-MB-231 and PC3 cells to perform experiments that require accurate cell number quantification without the inconvenience of dissociating spheroids by trypsinization. MDA-MB-231 and PC3 cells were grown to 60–70% confluency in 10 cm Petri dishes, trypsinized, and resuspended in CM in ULAPs. Media were changed twice a week. Cell aliquots were taken at 1, 3, 5, 7, and 15 days. The cells were fixed and stained for SA-βG activity. Like floating spheroids, after day seven, the majority of the MDA-MB-231 and PC3 cells were intensely blue (Figure 2). Appendix A shows a clearer progression of the increase in the SA-βG activity over time (1, 3, 5, and 7 days) in MDA-MB-231 and PC3 cells.

In Figure 2, floating MDA-MB-231 and PC3 cells have been fixed prior to the β-gal staining, giving the impression of true spheroids. Without fixation, these structures are loose aggregates of cells. These cells do not stick together, and even after 7 or more days, they can be separated into single ones by gentle mechanical dissociation with a pipette. However, as can be observed at a higher magnification (20×) in Figure 4 from our previous work [10], neither PC3 nor MDA-MB-231 formed true spheroids. These cell lines are referred as “non-spheroid-forming” cancer cell lines. This result indicates that cell detachment per se increases the SA-βG activity and that the formation of complex 3D structures is not necessary for the induction of a Q/S phenotype.

Quantifying the number of SA-βG^+^ cells in spheroids is not an easy task: attempts to quantify SA-βG^+^ cells in spheroids have been carried out by others using spheroid cryosections [30] or the relative intensity of SA-βG staining [31]. However, these methods do not easily discriminate between the exact percentage of SA-βG^+^ cells in the sample, or the relative size of SA-βG^+^ cells compared to SA-βG^−^ cells. For these reasons, MDA-MB-231 and PC3 cells were used in most of the following experiments that required cell counting.

### 3.3. Proliferative Activity Is Reduced Under AICs

A key characteristic of Q/S cells is that they withdraw from the cell cycle and remain metabolically active for a prolonged period [16,32]. To assess the proliferative capacity of cells grown under AICs, the aliquots of MDA-MD-231 and PC cells were taken at regular intervals, and the number of viable cells (detected by their ability to exclude trypan blue) was counted. Figure 3 shows that the cell number increases by around 40–50% within the initial 1–2 days of AICs and then slowly decreases over time. The rapid increase is likely due to the presence of cells that have already passed the G_1_/S restriction point and are committed to divide [33]. After this initial increase, the cell number slowly decreased overtime, indicating that there is no active proliferation. However, these cells remained viable and metabolically active as they retained the ability to metabolize MTT (Appendix A). Altogether, the intense blue staining and lack of cell division with evidence of viability is consistent with a Q/S state.

### 3.4. Cells Grown Under AICs Escape the Q/S State in a Plastic Rather than Stochastic Manner

Quiescence could be considered a plastic process because quiescent cells quickly resume proliferation when conditions are favorable [11]. In contrast, senescence cells tend to remain non-proliferative for extended time even when conditions are favorable and resume proliferation at variable times in a stochastic manner, as we and others previously observed. For instance, surviving glioma cells treated with temozolomide, carmustine, and lomustine remained viable in a senescent state for prolonged times. Cells that survived 3 weeks of exposure to these alkylating drugs were still viable after 4 months when maintained in drug-free media [34]. However, these cells escaped the senescent state at variable times, as was observed in vitro in our previous studies [34,35]. To address the nature of the escape of SA-βG^+^ floating cells, SA-βG expression was induced in PC3 and MDA-MB-231 by cell detachment in the presence or absence of Eto (50 μM) for 7 days. This dose of Eto was selected based on previous dose–response experiments in which we determined the IC_50_ for Eto using an MTT assay (~2.5 µM) in an attached cell culture (unpublished data). Eto is a well-known inductor of senescence [36,37]. Appendix A shows increased expression of SA-βG in H460 and H23 by Eto 1.25 μM in cells growing under ADCs. Because cells growing under AICs are more resistant to conventional anticancer drugs, we increased Eto to 50 µM for floating cells. This concentration provided a good balance between the number of surviving cells and the induction of senescence for each cell line tested, which was essential for the purpose of this study. Viable floating SA-βG^+^ cells were allowed to reattach in drug-free media at a low density in regular 6-well plates or 35 mm Petri dishes. Eto-treated SA-βG^+^ cells rapidly reattached but did not immediately resume an active proliferative state. The majority of these cells remained SA-βG^+^ and viable for a prolonged period (Figure 4). Viability was confirmed at day seven post reattachment by detecting the ability of reattached cells to reduce MTT (Appendix A). In contrast, Eto-untreated SA-βG^+^ cells reattached as fast as Eto-treated SA-βG^+^ cells but became SA-βG^−^ within 24 h (Figure 4).

These cells quickly resumed an active proliferative state that was clearly observed by crystal violet staining (Figure 5), and they produced visible colonies faster than Eto-treated SA-βG^+^ cells. It is important to clarify that these experiments were conducted to test the ability of Q/S cells to reattach and proliferate as well as to evaluate the SA-βG activity upon reattachment in a qualitative rather than quantitative manner. We found that Eto-treated PC3 cells tend to form sticky aggregates that made it difficult to count and load an equal number of Eto-treated and untreated cells. As a result, in the picture shown in Figure 4, it looks like Eto-treated cells grow faster than untreated cells. We selected this image because it clearly shows that the SA-βG activity persists in Eto-treated cells.

Detailed microscopic observation of reattachment of spheroids showed that cells that contact the surface of the plate tend to lose the SA-βG signal faster than those that do not (Appendix A). This observation suggests that cell–extracellular matrix interaction rather than cell–cell contact is an important factor that modulates the expression of SA-βG and may explain its plastic expression: cellular detachment from the extracellular matrix may be the trigger for the increase in the expression of SA-βG that is maintained as long as the cells remain in the floating condition. Such plastic behavior was observed for the cell detachment chemoresistant phenotype and the plastic expression of key proteins of several cellular pathways previously reported by our lab [38]. A reversible SA-βG activity, consistent with a plastic quiescent state, was observed in cells induced by serum starvation or confluence [39]. Under ADCs, deprivation of serum for 5 days induced a seven-fold increase in SA-βG [16]. It is important to note that adherent cells growing under prolonged periods (7–12 days) of serum starvation become highly resistant to conventional anticancer drugs such as paclitaxel, hydroxyurea, and colchicine [40]. Thus, is it conceivable that a reversible chemoresistant phenotype and a reversible Q/S state are two associated processes, closely connected, that could be exploited to develop new therapeutic strategies. The present study does not attempt to discriminate between quiescence and senescence in floating cells. While this distinction is important, it will require a set of experiments beyond the scope of this study.

These results suggest that cell detachment per se triggers a plastic quiescent state rather than a senescent state.

### 3.5. Lactose Increases the Intensity of β-Gal Expression in Cells Grown Under AICs

From the natural substrate of the enzyme β-Gal, we investigated the effect of lactose on cell detachment-induced Q/S status. Lactose is a disaccharide found solely in milk and dairy products and is cleaved by intestinal lactase to produce galactose and glucose. Galactose is metabolized primarily by the Leloir pathway and converted into glucose (Figure 6).

The link between cancer and lactose is inconclusive and sometimes contradictory [41], in part because most of the studies evaluated the consumption of dairy products (as indicator of lactose consumption) instead of pure lactose.

All our cell culture experiments described so far were performed in RPMI-1640 media that lack lactose. To investigate the effect of lactose on a cell-detachment-induced Q/S state, floating cells generated as described for previous experiments were incubated with 100 μM lactose for seven days. This concentration was chosen because it is close to the highest physiological concentration (75 +/− 18 μM) found in lactating women [42]. Literature data have shown that lactose at a concentration >20 mM is an inductor of senescence in normal human fetal lung fibroblast MRC-5 cells. However, these studies have been performed in cells growing under ADCs and utilized a very high concentration of lactose [43].

We found that the addition of 100 μM lactose under AICs increased the intensity of the SA-βG staining after 7 days of incubation (Figure 7), providing a direct link between cellular Q/S status and lactose under more realistic conditions.

### 3.6. Dysregulation of Protein Expression Involved in Lactose Metabolism Is Associated with Poor Prognosis in Breast Cancer

While some studies suggest a potential link between dairy consumption and certain cancers, particularly liver and breast cancer, our preclinical data clearly suggests that lactose, by inducing a Q/S status, may have an impact on the survival of cancer patients. To evaluate the impact of lactose metabolism on cancer survival, we conducted a bioinformatic analysis. We first conducted a differential gene expression analysis (DGE) as described in the Materials and Methods section. Of the original query 306 genes, 223 unique carbohydrate metabolism pathway/Leloir pathway genes were found among the top 1000 most significantly differentially expressed genes in the TCGA BRCA dataset (see Appendix A). The expression levels of the 223 genes found by the DGE were correlated with breast cancer patient prognosis, resulting in 23 unique carbohydrate metabolism pathway/Leloir pathway genes that were significantly correlated with survival (see Table 1).

These included two important genes from the galactose metabolism pathway: GALK1, a key enzyme of the Leloir pathway that phosphorylates galactose to galactose-1-phosphate and 2 (UGP2), the only enzyme capable of converting glucose 1-phosphate to UDP-glucose in mammalian cells [44]. Of the 23 total genes found to have statistically significant correlations with breast cancer survival, 17 genes were associated with a poorer patient prognosis when over-expressed. The other six genes were associated with lower survival when under-expressed (Appendix A). From the clinical point of view, over-expression is the same as high expression, and under-expression is the same as low expression. In the bioinformatics analyses, a patient’s gene expression level (for a given gene) is considered high (i.e., over-expressed) if their normalized expression value is greater than the median expression level (for that given gene) across all patients in the dataset. Likewise, a patient’s gene expression level (for a given gene) is considered low (i.e., under-expressed) if their normalized expression value is less than the median expression level.

Kaplan–Meier plots indicated that, for both galactose pathway genes, over-expression was significantly associated with a poorer prognosis in breast cancer patients (see Figure 8) (*p* = 0.024 for UGP2, *p* = 0.008 for GALK1).

Our bioinformatic analysis is in agreement with previous studies that linked the expression of critical genes of the galactose metabolism pathway (including UGP2, GALK1) with poor disease-free survival in a cohort of young breast cancer patients [45]. GALK1 could be a biomarker for predicting breast cancer metastasis [46]. On the other hand, UGP2 was found to be dysregulated in several cancers. High UGP2 expression was correlated with an increased rate of progression and poor prognosis in pancreatic cancer [44]. In hepatocellular carcinoma (HCC), low UGP2 expression levels were associated with tumor progression and a poor prognosis [47]. Our study, limited to breast, prostate, and lung cancer, did not find any association between GALM and cancer survival. However, high expression of GALM was associated with a worse prognosis in gliomas [48].

## 4. Conclusions

This study reported that cell detachment triggers a plastic Q/S-like state associated with a reversible increase in the SA-βG marker that can be further increased by high but physiologically relevant concentrations of lactose. In addition, bioinformatic analysis and a review of the literature data indicate that dysregulation of key enzymes of galactose metabolism including the Leloir pathway is associated in a tumor type-dependent manner with cancer prognosis. Finally, our data warrant further studies on the metabolism of galactose and its association with a Q/S status, which may unravel mechanistic insights of lactose consumption and cancer prognosis.

## Figures and Tables

**Figure 1 cells-14-01667-f001:**
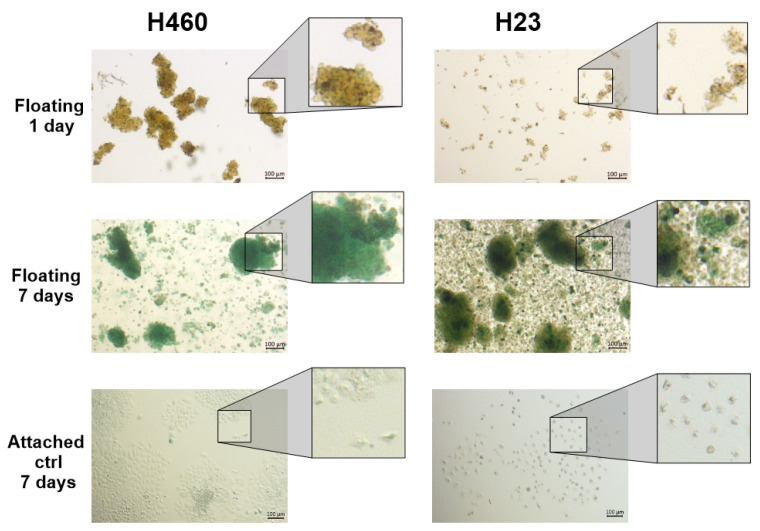
Spheroid-forming cancer cells growing under anchorage-independent conditions express high levels of SA-βG activity. H460, and H23 cells grown for 7 days under AICs (floating) were collected at the indicated times, fixed, and stained for SA-βG activity. Images were taken at 5× magnification. H460, and H23 cells grown for 7 days under ADCs (attached) were used as a control.

**Figure 2 cells-14-01667-f002:**
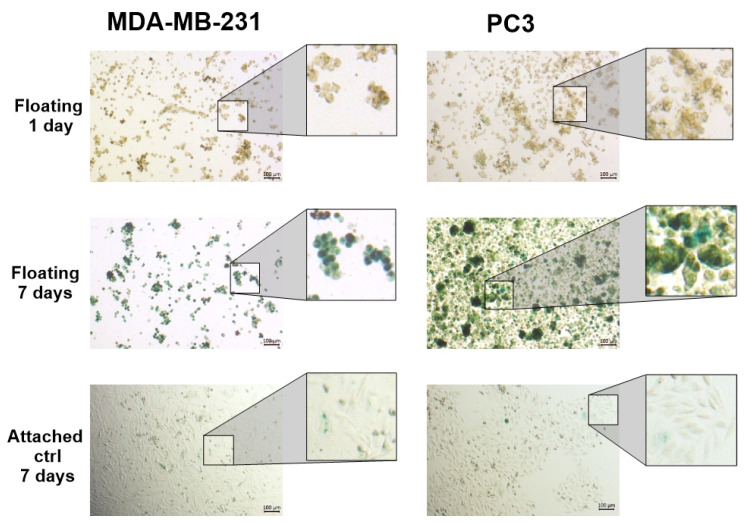
Non-spheroids forming cancer cells growing under anchorage-independent conditions express SA-βG activity. MDA-MB-231 and PC3 cells grown for 7 days under AICs were collected at the indicated times, fixed, and stained for SA-βG activity. Images were taken at 5× magnification.

**Figure 3 cells-14-01667-f003:**
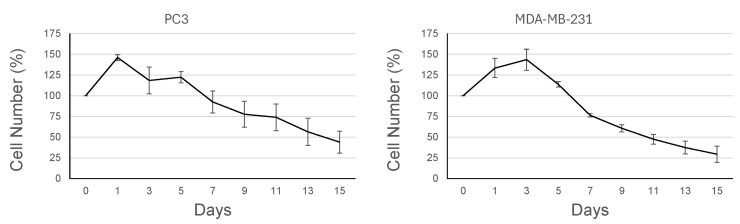
Cancer cells growing under anchorage-independent conditions stop proliferating. MDA-MB-231 and PC3 cells grown as adherent monolayers were harvested by trypsinization and transferred to ULAPs (Day 0 = 100%). Cell number was counted and expressed as % of its initial number at the indicated times (1, 3, 5, 7, 9, 11, 13, and 15 days). Data are mean ± SE of three independent experiments.

**Figure 4 cells-14-01667-f004:**
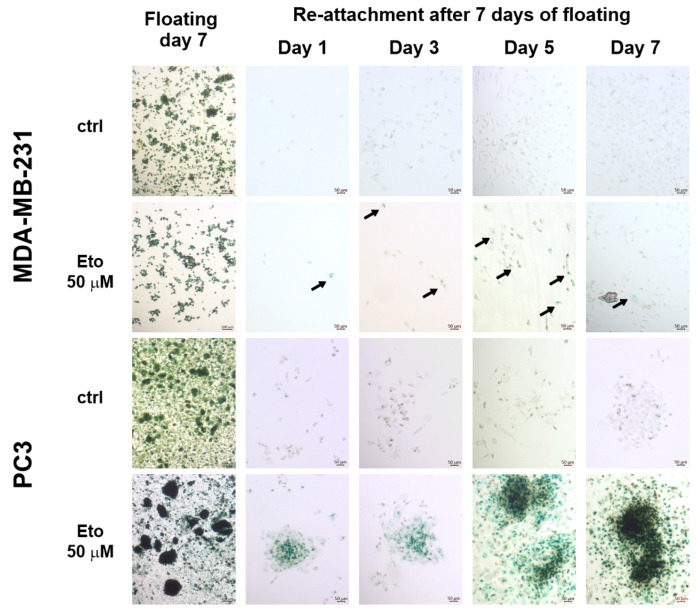
Reattachment induces a faster decrease in beta gal expression in detachment-induced senescent cells compared to Eto-induced senescent cells. MDA-MB-231 and PC3 cells grown under AICs for seven days in the absence or presence of Eto were allowed to reattach for 7 days. Samples were collected at the indicated time and stained for SA-βG activity. Arrows indicate SA-βG^+^ cells in Eto-treated cells. Images were taken at 5× magnification.

**Figure 5 cells-14-01667-f005:**
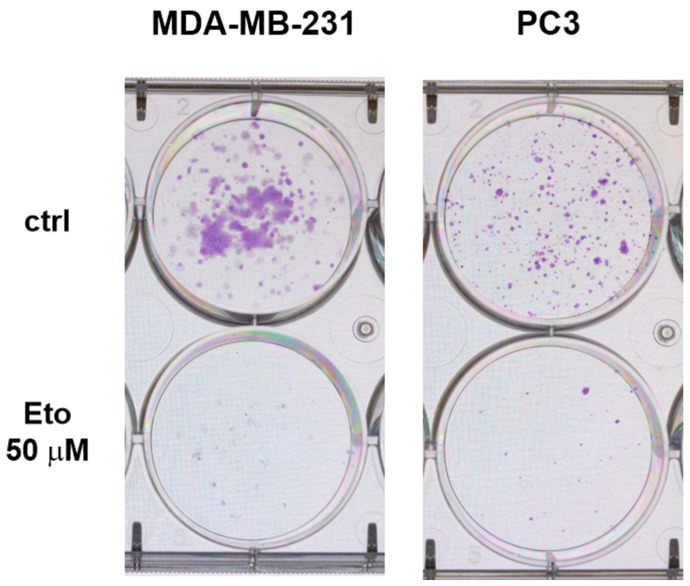
Detachment-induced senescent cells resume proliferation faster than Eto-induced senescent cells. MDA-MB-231 and PC3 cells grown under AICs (top wells) or treated with Eto (50 μM) (bottom wells) for seven days were allowed to reattach and grow in 6-well plates. Crystal violet staining was performed after eleven days.

**Figure 6 cells-14-01667-f006:**
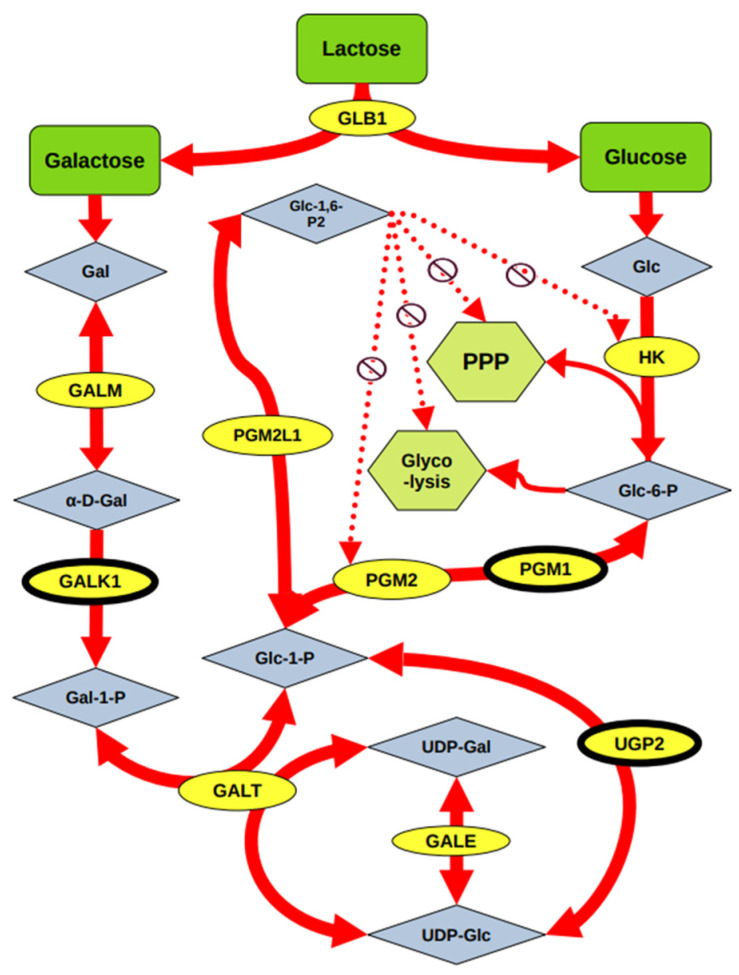
Schematic showing the roles of specific Leloir pathway enzymes that facilitate galactose metabolism. Lactose can be converted to galactose, which in turn can be converted to glucose via multiple possible paths. GALM and GALK1 activity convert galactose into galactose-1-phosphate, which is then metabolized by GALT to glucose-1-phosphate. Glucose-1-phosphate can either be converted via PGM1 and PGM2 to glucose-6-phosphate (which can then enter the pentose phosphate pathway (PPP) and glycolysis pathway), or it can be metabolized via PGM2L1 to glucose-1,6-P2, which acts as an allosteric inhibitor of enzymes HK and PGM2, thereby also inhibiting PPP and glycolysis. Dotted arrows indicate negative feedback regulation. Enzymes GALK1 and UGP2 are highlighted in the schematic with heavy black outlines.

**Figure 7 cells-14-01667-f007:**
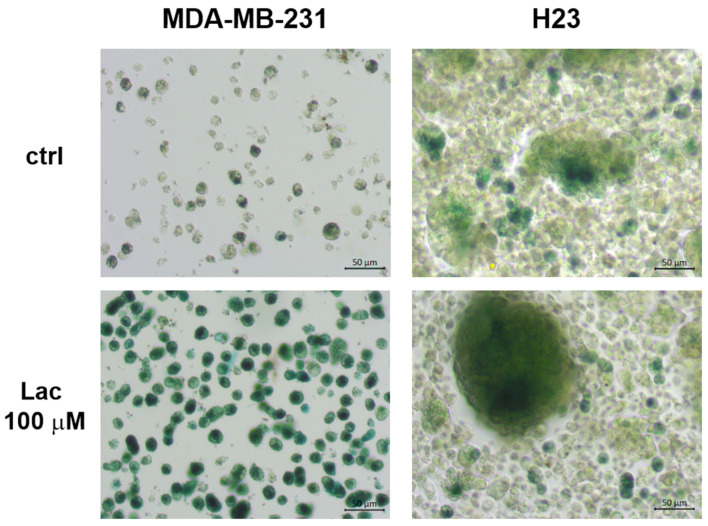
Lactose increases the expression of SA-βG staining. MDA-MB-231 and H23 cells were grown under AICs for 7 days in the absence (**top** panels) or presence of 100 μM lactose (**bottom** panels).

**Figure 8 cells-14-01667-f008:**
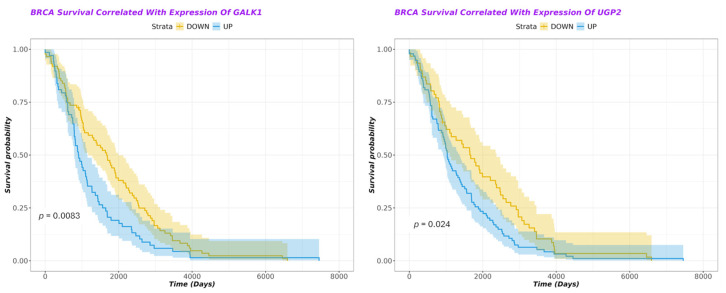
Kaplan–Meier survival plots of galactose pathway gene expression correlated with breast cancer patient survival. Blue line is over-expression, orange line is under-expression. Shaded regions around lines indicate confidence intervals. Over-expression of both GALK1 (on **left**) and UGP2 (on **right**) is significantly associated with worse patient prognosis (*p* < 0.05).

**Table 1 cells-14-01667-t001:** Summary of carbohydrate metabolism pathway/Leloir pathway genes significantly correlated with BRCA survival prognosis.

Gene Symbol	Pathway	Expression	LogFC	Effect on Survival	*p*-Value (Survival)
ACOT8	Dicarboxylate metabolism	DOWN	−2.3	Deleterious	0.00012
GSTO2	Ascorbate metabolism	DOWN	−1.9	Deleterious	0.00099
IDH1	TCA pathway	DOWN	−2.3	Deleterious	0.0024
KYAT3	Dicarboxylate metabolism	UP	0.95	Improved	0.0042
ALDOC	Glycolytic processes	UP	0.77	Improved	0.0056
HK3	Glycolytic processes	UP	0.56	Deleterious	0.0059
DLST	TCA pathway	DOWN	−0.42	Deleterious	0.0063
ADHFE1	Dicarboxylate metabolism	UP	0.7	Deleterious	0.0067
GALK1	Galactose metabolism	UP	0.6	Deleterious	0.0083
QPRT	Dicarboxylate metabolism	UP	0.8	Deleterious	0.0087
PFKL	Glycolytic processes	UP	0.41	Deleterious	0.017
SLC37A4	Gluconeogenesis pathway	UP	0.5	Improved	0.017
IDH3B	TCA pathway	UP	1.1	Deleterious	0.019
UGP2	Galactose metabolism	DOWN	−1.03	Improved	0.024
CSKMT	TCA cycle pathway	UP	0.35	Improved	0.027
AGXT	Glyoxylate metabolism	DOWN	−0.43	Improved	0.029
ASL	Dicarboxylate metabolism	UP	0.81	Improved	0.029
PGK1	Glycolytic processes	UP	0.35	Deleterious	0.03
MTHFD1L	Dicarboxylate metabolism	DOWN	−1.2	Improved	0.03
PGAM1	Glycolytic processes	UP	0.39	Deleterious	0.032
UGT2B15	Pentose glucuronate interconversions	UP	0.15	Improved	0.034
DHTKD1	Glycolytic processes	DOWN	−0.16	Deleterious	0.037
GAA	Sucrose metabolism	UP	0.28	Improved	0.049

## Data Availability

The data that support the findings of this study are available from the corresponding author upon reasonable request.

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
