# Peer review of "Reversible Upregulation of the Senescence-Associated Beta-Galactosidase Marker Induced by Cell Detachment in Cancer Cells"

_cells, 2025, doi:10.3390/cells14211667_

Round 1

Reviewer 1 Report

Comments and Suggestions for Authors

cells-3796797-peer-review-v1
The work is in some way challenging because the role of cellular senescence in cancer resistance and metastasis is difficult to crack – it is a two-edged sword (Xiao  DOI: 10.3389/fonc.2023.1189015), the authors also mention the reversibility of senescence in cancer associated with reprogramming to stemness [refs 14,15].
The working hypothesis:
“During metastasis, cancer cells detach from the primary tumor and the floating cells enter the circulation and reattach in distant organs. Floating cells are highly chemoresistant to anticancer drugs, but the underlying mechanisms are poorly understood. We hypothesized that floating cells transition into a quiescent/senescent (Q/S) state”.
The authors took the well known as aggressive cancer cell lines - two non-small cell lung cancinoma, one TNBC (MDA MB 231), and androgen-resistant prostate cancer PC3 – all highly mutant and aneuploidy (para-triploid) and incubated them in the plates covered with special plastic which does not allow cells to attach (AICs). Tumour cells formed spheroid-like clusters but two of them (MDA MB 231 and PC3) did not, however all attained senescence, judged here by the positively staining by Sa-b-gal pH6 reaction. Incubation with Etoposide increased this reaction. When allowed to attach on slides for the next 7 days, the cells proliferated and nearly lost senescence marker; after ETO treatment, they also attained senescence but faster recovered, when attached. 
These findings do not cause any surprise, as to some extent, they are not novel – senescence induction in spheroid cultures of mesenchymal stem (Rovere DOI: 10.3389/fbioe.2023.1297644)  and also tumour cell lines has been described (Lee, HG., Kim, J.H., Sun, W. et al. Senescent tumour cells building three-dimensional tumor clusters. Sci Rep 8, 10503 (2018). https://doi.org/10.1038/s41598-018-28963-0).
Two cell lines which did not form clusters, MDA MB 231 and PC3 (both mt TP53 and KRAS-mutant) after 7 days growth in AICs were disseminated and remained such in the attachment favouring slides, so the contact inhibition was absent in them in any conditions.
What is causing a surprise: 
(1)The authors’ assumption that the floating huge spheroids in non-attaching plates/wells, which in fact stand at place (or maybe, fluctuate) are the same as migrating tumour cells. Spheroids are not the same as tumour cells circulating in blood and producing metastases in distant tissues. To really move in blood, the cell surface must possess the modified peripheral cytoskeleton enabling “shear flow” – really swim and migrate in narrow lymph and blood vessels. They can gain this property by association with macrophages (doi: 10.1073/pnas.1320198111) or adapting themselves by mesenchymal-amoeboid transition (MAT), along with the microenvironment, as known from the literature (doi.org/10.1016/j.bbadis.2024.167332).
•    Whether their surface becomes modified in spheroids in this way, whether lactose favours MAT – needs to be shown or cited from the literature, which says that MAT is energetically most beneficial: in favour of aerobic glycolysis, MAT suppresses oxidative phosphorylation to the minimum (DOI: 
•    10.1016/j.tcb.2022.09.009). 

(2) The appearance of the Sa-b-gal-positive cells floating under AICs conditions for 7 days presented by the authors, in control and when incubated with lactose. We see here on the cultures which cluster and do not (Fig.7) the Sa-b-gal stains not perinuclear cytoplasm as due, but preferentially cell nuclei and does it more violently - in the presence of lactose. However, in the surface-anchored cultures, the Sa-b-gal staining is convential, cytoplasmic and blue, seen though all figures, also in the Supplemental Figures.  
This difference in the localisation of the staining product was not described and commented by the authors. 
In Methods, the authors refer to the use of the original method by Dimri [ref 18] but their protocol differs from that by the prolonged fixation by PF – 30 min, while Dimri recommended only 3-5 min.
I found the protocol used by Judith Campisi’ lab, who worked much on cell senescence and there is her warning: “DO NOT OVERFIX – 5 MIN MAXIMUM; REDUCE TIME IF STAINING IS WEAK”. They also recommend the fixation by PG combined with GA (a variant also suggested by Dimri), as preserving morphology much better. 
Sa-b-gal reaction at pH 6 is known to be due to activity of the lysosomes in senescent cells, which acidify the cytoplasm. From the view of senescence mechanism, the shown nuclear staining in the AICs conditions is an artefact.
So, what still can be stained for galactosidase in cell nuclei here? 
There are the data of the alternative splicing forms of the galactosidase gene, which is associated with regulation of transcription and have the nuclear location. Its origin is from E. Coli and yeast and may be due to recombinative hybrids, etc (https://doi.org/10.1016/0092-8674(84)90055-2).
Galactosidase alternative splicing may be associated with the nerve tumour gene of neurofibromatosis (mutant NF1 in MDA MB 231), be involved in brain development or induced through FOS gene by stress (doi: 10.1371/journal.pone.0107238). I did not dig all this literature – the authors can find it. This issue is rather obscure as related to mammalian cells.
Anyway, this non-convential for senescence nuclear staining by Sa-b-gal pH6 in the floating cells, clustered or un-clustered compromises the “floating senescence” hypothesis proposed by the authors. At the same time, in spite of the absence of contact inhibition even in the attached cancer cultures, they acquire the conditional pattern of Sa-b-gal staining (in the same hands and method?) –a light-blue perinuclear mark. This difference, although hardly seen because of poor morphology, needs explanation.
(3) I also regret that the authors did not notice polyploidy in ETO-treated cancer cells (it should be there), while the resolution of images is rather poor. I would remind the seminal article by Weihua (2011, doi: 10.1002/cncr.26021 ) , that single giant cancer cells that are also Sa-b-gal-positive can initiate metastatic tumours when injected in vivo. Circulating aneuploidy PC3 prostate cancer (cisplatin-treated) are pro-metastatic and may be dormant (Mallin et al doi: 10.1158/1541-7786.MCR-24-0689). The role of circulating giant cells for poor patient prognosis has been also shown for some other types of tumours as published. The therapy-induced senescence interconnection with polyploidy of cancer cells was noticed 15-20 years ago and confirmed (Sikora et al 2020 doi.org/10.1016/j.semcancer.2020.11.015).
At the same time, the idea of the Lenoir galactose circuit supported here by the bioinformatics analysis of TCGA breast cancer tumours vs normal cells, perfomed in String and KEGG systems, its possible role for the energetic of the non-adherent cancer cells, the scheme, the Kaplan-Meir curves – all this is OK and is valuable by itself, unclosing another facet of cellular senescence, for poor patient prognosis. The authors could try to look for the correlation with Rho-kinases (cdc42 and RAC1) modifying actin for MAT, to support their hypothesis from mechanic point of view, and add the modern senescence gene bioinformatics platform (e.g. https://maayanlab.cloud/Harmonizome/gene_set/cellular+senescence/GO+Biological+Process+Annotations+2015) – it contains p21/CIP1 and p16INK4a – the latter - an p53-independent marker for reversible senescence associated with reprogramming and also IL6 (Moistero et al., 2021 doi: 10.1111/acel.12711), which are more reliable than the tricky Sa-b-gal staining.  If they are interested in the accompanying polyploidy (and they should because this subpopulation will canalize the properties of migrating polyploid cells for the principal component analysis with a few top genes), they could add testing this BRCA database with the Absolute algorithm for the whole genome duplications (https://doi.org/10.1016/j.ccell.2018.03.007), to find the possible correlations with Lenoir pathway and MAT markers. I do not doubt in the positive result if performed correctly. It could be a good contribution in the puzzle of cancer cell senescence and form an independent article. You can start from your publication [ref10], which logically leads to it. 
However, in its current presentation, the article is not well updated for the literature context, its working hypothesis is only partly supported; its experimental design is ambiguous, and therefore it cannot be recommended for publication. 

Reviewer 2 Report

Comments and Suggestions for Authors

Overall, this study presents some interesting observations. However, the study is disjointed and doesn’t present a single coherent story. In the initial experiments described the authors have focused on anchorage independent growth and state clearly that the Breast cancer cells MDA-MB-231 and prostate cells PC3 do not form spheroids in these conditions. It is, therefore, not clear why has breast cancer been focused on for the differential gene expression analysis when the MDA-MB-231 cells were the negative control in the study? Furthermore, there is extensive literature showing that the Cancer Stem Cell population of MDA-MB-231 do form spheroids when grown under specific conditions.

Specific concerns the authors need to address:

  • Line 53 - not sure this sentence makes sense
  • Could the authors comment on the appropriateness of this culture system to replicate the conditions detached cancer cells grow in? Would the use of stem cell media be more appropriate which is commonly used for growing Cancer Stem Cells which commonly grow as spheroids in anchorage independent conditions?
  • How was the dose of etoposide selected for this study?
  • Figure 1- These don't look like spheroids. Why are at 1 day the cells should still be small clumps if they have been established from single cells. This looks like the cells have not been made into single cells before they were plated onto the ULAPs.
  • Line 176- Not scientific language
  • Figure 2- Some of these structures especially in the PC3 cells look more like spheroids than in the cell lines that are meant to form spheroids.
  • Is it possible to quantify the level of MTT staining as is usually presented?
  • Was an etoposide treated adherent control included if this agent is being used as it is able to induce senescence?
  • The PC3 cells treated with etoposide are clearly growing more than the untreated control cells. I agree that the MDA-MB-231 seem to be growing less and are retaining the B-gal staining. Also, the PC3 cells aren't comparing like for like regarding cell density.
  • Could the day 0 or day 1 controls be included to show that the same number of cells were seeded at the star of the experiment
  • Fix the typos - there are numerous places in the manuscript where there are additional spaces present in the text
  • Figure 8 how do you know if it is over or under expression. In patients it is usually described as high or low expression

Reviewer 3 Report

Comments and Suggestions for Authors

The current version lacks the mechanistic and biological depth expected for publication

Reviewer 4 Report

Comments and Suggestions for Authors

The results presented in this article are concerning exploring the hypothesis that cell detachment is a reversible quiescent/senescent-like (Q/S-like) state that may contribute to the chemo resistance of cancer cells growing under AICs.

It was shown that during metastasis, cancer cells detach from the primary tumor, and the floating cells enter the circulation and reattach in distant organs. They are highly chemo resistant to anticancer drugs. Authors was also hypothesized that floating cells transition into a quiescent/senescent (Q/S) state.

It was also observed that the human lung carcinoma H460 and H23, human prostate adenocarcinoma PC3, and human breast adenocarcinoma MDA-MB-231 cells were growing under anchorage-independent conditions. Authors detected increase of B-Galactosidase activity, marker associated with cell proliferation. Similar experiments were performed with cells treated with etoposide (Eto), a inductor of senescence. Eto-untreated floating cells resumed proliferation faster and showed a quicker decrease in β-Galactosidase activity compared to Eto-induced  senescent cells. Authors find partial answer for the question concerning chemo resistance under anchorage-independent conditions and a new target to eliminate highly resistant floating cells.  

They conclude that cell detachment per se triggers a reversible (plastic) increase of β-Galactosidase. Authors findings provide a partial explanation for chemo resistance under anchorage-independent conditions and a new target to eliminate highly resistant floating cells. Ultimately, eliminating Q/S floating cells may prevent or reduce metastasis.

Round 2

Reviewer 2 Report

Comments and Suggestions for Authors

Thank you for the detailed responses to my comments, you have addressed most of my concerns. However, several points in your response need to also be included in the main text of the manuscript. 

Specifically, the authors have still not added to the text why/how the etoposide dose was selected.

The response to comment 5/7 was clear and it would be useful if the authors add the definition they are using to determine whether a cell line is able to form a 'true spheroid' to the main text.

I agree with the authors that etoposide is a well-documented inducer of senescence in adherent cells, however, the inclusion of this data serves as a control for their floating cells in this case is not novel information but should still be included. 

In the bioinformatics analysis they authors need to include the clarification they have provided in their response to my comments regarding how they determined whether the expression was high or low. 

Author Response

Find attached.
